# Comparative Metabolite Profiling of Traditional and Commercial Vinegars in Korea

**DOI:** 10.3390/metabo11080478

**Published:** 2021-07-24

**Authors:** Minhye Shin, Jeong-Won Kim, Bonbin Gu, Sooah Kim, Hojin Kim, Won-Chan Kim, Mee-Ryung Lee, Soo-Rin Kim

**Affiliations:** 1Department of Microbiology, College of Medicine, Inha University, Incheon 22212, Korea; alsgp01@gmail.com; 2School of Food Science and Biotechnology, Kyungpook National University, Daegu 41566, Korea; jungwon1103@naver.com (J.-W.K.); ant369369@naver.com (B.G.); 3Department of Environment Science & Biotechnology, Jeonju University, Jeonju 55069, Korea; skim366@jj.ac.kr; 4Experimental Research Institute, National Agricultural Products Quality Management Service, Gimcheon-si 39660, Korea; rex7878@korea.kr; 5School of Applied Biosciences, Kyungpook National University, Daegu 41566, Korea; kwc@knu.ac.kr; 6Department of Food and Nutrition, Daegu University, Gyeongsan 38453, Korea

**Keywords:** GC/MS, metabolomics, vinegar, lactic acid, propanoic acid, erythritol, 2,3-butanediol

## Abstract

Vinegar, composed of various organic acids, amino acids, and volatile compounds, has been newly recognized as a functional food with health benefits. Vinegar is produced through alcoholic fermentation of various raw materials followed by acetic acid fermentation, and detailed processes greatly vary between different vinegar products. This study performed metabolite profiling of various vinegar products using gas chromatography–mass spectrometry to identify metabolites that are specific to vinegar production processes. In particular, seven traditional vinegars that underwent spontaneous and slow alcoholic and acetic acid fermentations were compared to four commercial vinegars that were produced through fast acetic acid fermentation using distilled ethanol. A total of 102 volatile and 78 nonvolatile compounds were detected, and the principal component analysis of metabolites clearly distinguished between the traditional and commercial vinegars. Ten metabolites were identified as specific or significantly different compounds depending on vinegar production processes, most of which had originated from complex microbial metabolism during traditional vinegar fermentation. These process-specific compounds of vinegars may serve as potential biomarkers for fermentation process controls as well as authenticity and quality evaluation.

## 1. Introduction

Vinegar is currently recognized as a functional food due to its potential health benefits, including antioxidant [1], antidiabetic [2], cholesterol-lowering [3], weight-reducing [4], and immune-boosting [5] activities. The most representative vinegars include balsamic, apple, and brown rice vinegars, and their authentic traditional products are preferred by consumers [6]. Recent metabolomic analysis of traditional vinegars suggested that vinegar fermentation processes alter vinegar compounds and possibly their functional properties [7,8,9,10,11,12].

Traditional vinegar is produced by two consecutive fermentation processes: alcoholic fermentation and acetic acid fermentation. During alcoholic fermentation, yeast (*Saccharomyces cerevisiae*) converts fermentable sugars in the raw materials into ethanol. Then, acetic acid bacteria are introduced and ethanol is oxidized into acetic acid [13]. Traditional vinegar fermentation is often performed spontaneously by indigenous microorganisms, leading to a slow and complex fermentation process [14]. On the other hand, most of the commercially available vinegars are produced by fast acetic acid fermentation of distilled ethanol by a starter culture (the mother of vinegar) [15].

Vinegar is composed of various organic acids, amino acids, and volatile compounds that originate from raw materials and are produced by microbial fermentation. Different metabolite profiles of vinegar contribute to different flavors and functional properties [16,17]. Previously, non-volatile metabolites of Korean traditional vinegars were compared to those of commercial vinegars, and the presence and levels of some sugars and acids were identified as being significantly different between the two groups [18]. However, the unique volatile metabolites of Korean traditional vinegars have rarely been investigated. 

In this study, volatile and non-volatile metabolite profiling of traditional and commercial vinegars in Korea was performed using gas chromatography–mass spectrometry. Specifically, all seven traditional vinegars certified so far in Korea (the Ministry of Agriculture, Food and Rural Affairs, Korea) were used for the comparative metabolite profiling. Additionally, vinegars made from various ingredients were included in both groups (traditional vs. commercial) to focus on different fermentation processes rather than raw materials. Through this comparative metabolite profiling, the purpose of this study is to identify vinegar compounds specific to traditional fermentation processes.

## 2. Results and Discussion

### 2.1. Metabolomic Differences between Traditional Vinegars and Commercial Vinegars

From the GC/MS analysis of seven traditional vinegar (TV) and four commercial vinegar (CV) samples (Table 1), 102 volatile and 78 nonvolatile compounds were identified. The partial least square discriminant analysis (PLS-DA) of the volatile and nonvolatile metabolites separated the compounds by the TV and CV groups, as shown in the score plots (Figure 1). The distribution of the vinegar samples on the score plots suggested greater diversity in metabolites of the TVs than those of the CVs. The variable importance in projection (VIP) scores of the metabolites were positively correlated with the −Log (*p*-value) for the volatile and nonvolatile compounds (Figure 1B,D); the correlation was stronger for the volatile compounds. This result suggests that the metabolomic differences between the TVs and CVs are better characterized by the volatile compounds. Although the comparison of TVs and CVs for their non-volatile and volatile compounds was conducted for the first time in this study, some recent studies have also pointed out that aroma profiles of vinegars are critically changed by different fermentation processes as well as raw materials [9,19]. 

A total of 180 metabolites were identified from the TVs and CVs (Table 2, Appendix A Appendix A). Twenty of these metabolites were classified as being significantly different between the two groups (VIP scores > 1.0, PLS-DA; *p* < 0.05, *t*-test); among them, 18 metabolites were significantly more abundant in traditional vinegars, five of which were only detected in traditional vinegars (TV-only). On the other hand, two metabolites (ethyl 3-(methylthio)propionate and asparagine) were detected at significantly higher levels in commercial vinegars, and the former was only detected in commercial vinegars (CV-only). Eighteen out of 20 of the most significantly different metabolites were more abundant in the TV samples, suggesting that traditional vinegars were more enriched by both volatile and nonvolatile compounds. 

Vinegar metabolites are enriched during fermentation and aging processes. Recently, it was found that the contents of some metabolites of vinegars such as lactic acid increased mostly during alcoholic fermentation [7]. In addition, aroma profiling of Chinese vinegars using GC/MS and GC-O suggested that volatile compounds are enriched during vinegar aging [10]. Similarly, an NMR-based metabolomic study reported that the concentrations of most vinegar metabolites increased after aging longer than a year [20]. Because TVs undergo alcoholic fermentation and require longer fermentation and aging periods, compounds that are significantly more abundant in TVs would be closely associated with fermentation processes. Meanwhile, in a prior study comparing non-volatile compounds of TVs and CVs, various sugars were significantly more abundant in CVs [18]. However, those sugars were not representative compounds of CVs in the present study due to large variations. 

### 2.2. TV-Only Compounds

The five TV-only compounds only detected in traditional vinegars were 2,4,5-trimethyl-1,3-dioxolane, 2-methoxyethyl acetate, phenylalanine ethyl ester, L-isoleucine, and phenylethylamine (VIP > 1.0 and *p* < 0.05) (Figure 2). Some of the compounds had wide interquartile and error ranges, suggesting that their contents varied greatly among different TV samples. Except for phenylalanine ethyl ester, four of the TV-only compounds were detected in all the traditional vinegar samples.

Three TV-only volatile compounds are unique metabolites associated with specific fermentation conditions. 2,4,5-Trimethyl-1,3-dioxolane was one of the compounds not found after fermentation but formed and accumulated during the aging process [10]. 2-Methoxyethyl acetate (acetic acid, methoxy-, ethyl ester) compounds are reported as flavor compounds of several ethnic foods, such as Korean soybean paste [21], Chinese liquor [22], and Burundian cassava flour [23]. Notably, 2-methoxyethyl acetate is detected only at the end of soybean fermentation; thus, it is associated with complex fermentation processes [21]. Phenylalanine ethyl ester (Phe-EE), one of the minor amino acid ethyl esters of wines, is formed during the second half of the fermentation when ethanol concentration is high [24,25].

Surprisingly, common food metabolites like L-isoleucine, an amino acid, and phenylethylamine, an amino acid derivative, were identified as nonvolatile TV-only compounds. L-isoleucine is one of the amino acids found in most traditionally fermented vinegars [6]. L-isoleucine and other branched-chain amino acids are associated with the production of higher alcohols and esters during fermentation [26]. Phenylethylamine is one of the least common biogenic amines in wine and dairy products [27,28]. Previously, phenylethylamine was not detected in all the vinegars tested [17,29].

Meanwhile, there were 51 other TV-only compounds, 43 volatile and eight nonvolatile compounds, with a low significance (*p* > 0.05) due to sample variations, which were independent from raw materials (Appendix A Appendix A). Because some of those compounds are strongly related to fermentation, a selected group of TV-specific compounds might be useful as a potential marker for traditional vinegars. For example, 3-phenyllactic acid (*p* < 0.085), one of the aromatic hydroxy acids that contributes to vinegar’s flavor, is produced during fermentation using phenylalanine as a precursor, mainly by lactic acid bacteria [8].

### 2.3. CV-Only Compounds

Ethyl 3-(methylthio)propionate was the significant CV-only volatile compound detected from two commercial vinegar samples (VIP > 1.0 and *p* < 0.05) (Figure 2). Ethyl 3-(methylthio)propionate has recently been identified as a flavor-active compound in Chinese vinegar and Chinese liquor [9,30]. Previously, ethyl 3-methylthiopropionate was detected only when sugars or spices were added during vinegar fermentation [9]. This observation is consistent with our conclusion that it is specific to CVs containing various additives.

There were 19 other CV-only volatile compounds with a low significance (*p* > 0.05) due to sample variations (Appendix A). The unique compounds of each sample might be associated with the different ingredients, such as brown rice, apple, and pineapple, of the CVs. For example, methyl 5-hexenoate, a unique volatile compound of pineapple [31], was detected only in CV2, a pineapple-based commercial vinegar. Notably, asparagine had the third-highest VIP score among the nonvolatile compounds due to its detection in all the CVs and only 1 TV.

### 2.4. Common Vinegar Compounds

Among the 20 compounds that were significantly different between TVs and CVs, four compounds (lactic acid, erythritol, propanoic acid, and 2,3-butanediol (denoted as its 2 TMS derivative, 3,6-dioxa-2,7-disilaoctane and hexamethyl, in some metabolomic studies)) were detected in all 11 vinegars tested; therefore, they were classified as common vinegar compounds (Figure 3). The four common vinegar compounds were nonvolatile and significantly high in TVs. Based on the median values of the common vinegar compounds, the relative abundance of lactic acid, 2,3-Butanediol, erythritol, and propionic acids were 71, 29, 9, and 4 times higher in TVs than CVs, respectively.

Various indigenous microorganisms are present during the fermentation of TV [32], thus explaining the high levels of lactic acid, erythritol, propanoic acid, and 2,3-butanediol in TVs. Specifically, indigenous lactic acid bacteria are the dominant bacterial population during the alcoholic fermentation stage of TVs [33]. Heterofermentative metabolism of lactic acid bacteria leads to the accumulation of lactic acid and other minor compounds, including propanoic acid and 2,3-Butanediol [34]. Additionally, erythritol is a byproduct of non-*Saccharomyces* osmotolerant yeast, such as *Candida magnoliae* and *Yarrowia lipolytica* [35,36], which predominates in the initial stage of static acetic acid fermentation [14,32]. During acetic acid fermentation, ethanol concentration is reduced to 6%–7%, allowing the growth of the non-*Saccharomyces* yeast such as *C. magnoliae*. Because most TVs are produced by static and slow acetic acid fermentation, the oxidation of erythritol to erythrulose by acetic acid bacteria may be limited [37], thus resulting in a high level of erythritol in TVs.

## 3. Materials and Methods

### 3.1. Raw Material Preparation

All 7 traditional vinegars certified so far in Korea (the Ministry of Agriculture, Food and Rural Affairs, Korea) were purchased directly from each manufacturer, and four representative commercial vinegars were purchased from local markets in Korea (Table 1). Vinegars with various raw materials were selected to offset different metabolites originating from different raw materials.

### 3.2. High-Performance Liquid Chromatography (HPLC) Analysis

Acetate, ethanol, glucose, and fructose (Sigma-Aldrich, St. Louis, MO, USA) were quantitated using an HPLC (1260 series, Agilent Technologies, Santa Clara, CA, USA) system with an RI detector and a Rezex-ROA Organic Acid H+ (8%, 150 × 4.6 mm) column (Phenomenex Inc., Torrance, CA, USA). The column was eluted with 0.005 N H_2_SO_4_ (Sigma-Aldrich, St. Louis, MO, USA) at 0.6 mL/min and 50 °C. The dynamic range of the standard curves was 0.1–5 g/L, and samples were appropriately diluted prior to analysis. All samples were analyzed in triplicate.

### 3.3. Sample Derivatization for Analysis

Vinegar samples were derivatized with some modifications to previously reported methods [38]. For methoxyamination, 50 μL of methoxyamine hydrochloride in 0.2% pyridine (*w/v*) (Sigma-Aldrich, St. Louis, MO, USA) was added to 5 μL of a dry-vacuumed vinegar sample. This mixture was then incubated for 90 min at 30 °C. For trimethylsilylation, 50 μL of N-methyl-N-(trimethylsilyl)trifluoroacetamide (Sigma-Aldrich) was added to the samples and incubated for 30 min at 37 °C.

### 3.4. Metabolite Analysis Using Gas Chromatography–Mass Spectrometry (GC/MS)

For volatile compounds, 5 mL of samples were extracted by SPME fiber coated with dinylbenzene/carboxen/polydimethylsiloxane (Bellefonte, PA, USA) at 50 °C for 60 min, as previously described [19]. The samples were injected by MPS autosampler (Gerstel, Muelheim, Germany) at a split mode (1:20) into the GC/MS system (7890-5977B; Agilent) equipped with a DB-WAS column (60 m × 0.25 mm, 0.25 μm thickness; Agilent). The initial oven temperature was set to 40 °C for 2 min, ramped up by 20 °C/min until reaching the final temperature of 240 °C, at which it was held for 5 min. The mass selective spectra were operated in scan mode with a mass range of 500–550 *m/z*. All samples were analyzed in triplicate.

For nonvolatile compounds, 1-µL aliquots of derivatized samples were injected in a split mode (1:50) into the gas chromatography–time-of-flight mass spectrometry (GC–TOF/MS) equipped with an Agilent 7890B GC, an RTX-5Sil MS column (30 m × 0.25 mm, 0.25-μm in thickness; Restek, Bellefonte, PA, USA), an integrated guard column (10 m × 0.25 mm, 0.25-μm in thickness; Restek), and a Pegasus HT TOF MS (Leco, St. Joseph, MI, USA). The initial oven temperature was set to 75 °C for 2 min, ramped up by 15 °C/min until the final temperature of 300 °C, and held for 1.5 min. The mass spectra of metabolites were obtained in the mass range of 45–500 *m/z* at an acquisition rate of 17 spectra/s. All samples were analyzed with six replicates. For laboratory quality control (QC), a blank and in-house QC mixture of alanine *2,3,3,3*-*d_4_*, glutamic-*2,3,3,4,4*-*d_5_* acid, asparagine-*^15^N_2_-d_8_* and xylitol (Sigma-Aldrich, St. Louis, MO, USA) at a concentration of 0.08 mg each were analyzed with each batch of analysis, and the coefficient of variation (CV) was kept below 10%.

### 3.5. Data Processing and Statistical Analysis

For data processing, raw peak area values were used, as previously described [8,18,39]. For the quality assurance of the data, CVs for measured metabolites was calculated, and it was confirmed that the 10 metabolites with the highest abundance of each biological group had CVs less than 10%. For identification of the metabolites, the raw data were processed by an automated mass spectral deconvolution and identification system (AMDIS) using the National Institute of Standard and Technology library (NIST 2014) with a similarity of more than 70% [40,41]. The pre-processed data were further processed by SpectConnect (http://spectconnect.mit.edu, accessed on 22 July 2021) [42] for peak alignment and for generating the data matrix. The statistical analysis was performed using partial least squares-discriminant analysis with STATISTICA (ver. 7.1; StatSoft, Tulsa, OK, USA) and MultiExperiment Viewer [43,44].

## 4. Conclusions

In this study, the metabolite profiles of TVs and CVs were analyzed and compared. A total of 180 volatile and nonvolatile compounds were identified, and TVs had a greater metabolite diversity than CVs. Twenty compounds significantly influenced the differentiation of TVs from CVs, which were classified into TV-only compounds, CV-only compounds, and common vinegar compounds. The compounds high in TVs were associated with complex microbial activities, while the compounds high in CVs were associated with commercial vinegar additives. The results provide comparative metabolite profiles of vinegars with different production processes. The selected compounds of vinegars can be applied as potential markers for vinegar fermentation processes as well as for authentic evaluation of vinegars in Korea.

## Figures and Tables

**Figure 1 metabolites-11-00478-f001:**
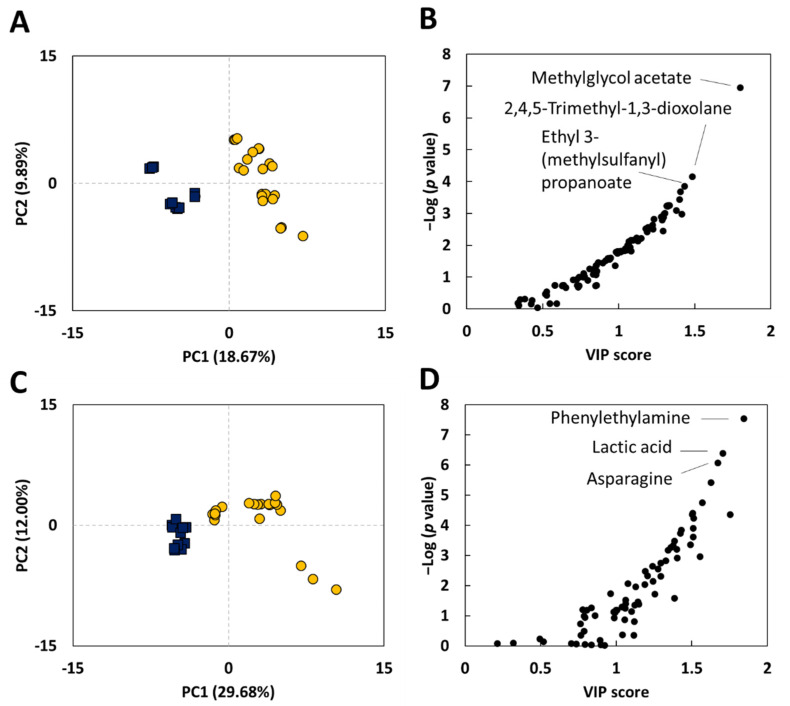
The partial least squares-discriminant analysis (PLS-DA) of the volatile (**A**,**B**) and nonvolatile (**C**,**D**) compounds of 11 vinegar samples (Table 1). (**A**,**C**) Score plots. (**B**,**D**) variable importance in projection (VIP) scores vs. −Log *p*-value plots of the 102 volatile (**B**) and 72 nonvolatile (**D**) compounds.

**Figure 2 metabolites-11-00478-f002:**
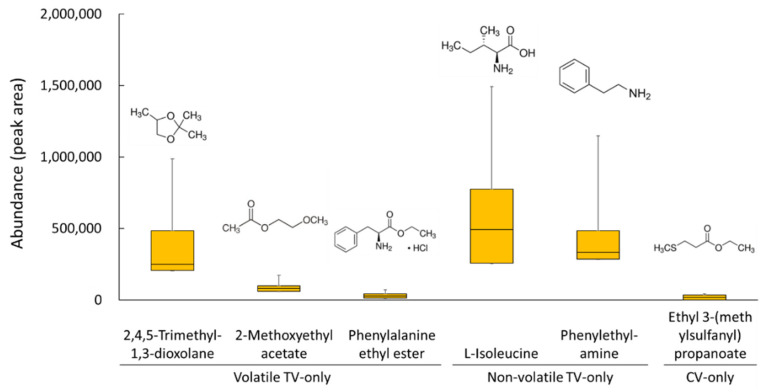
The abundance of six metabolites detected only in traditional vinegar (TV-only) or commercial vinegar (CV-only). The interquartile range (boxes), the largest and smallest values (error bars), and the medians (horizontal lines in the boxes) are indicated.

**Figure 3 metabolites-11-00478-f003:**
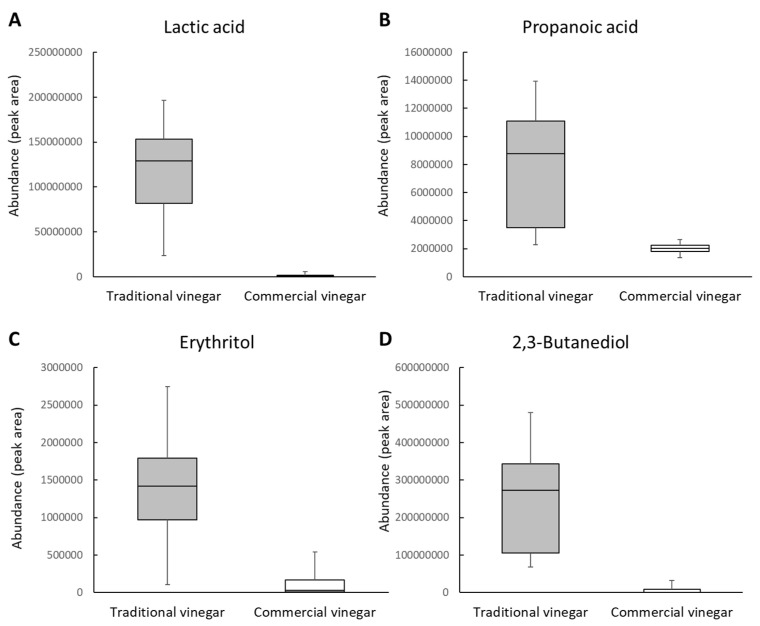
The comparison of the significantly more abundant compounds in TVs than those in CVs. Among the 20 significantly different compounds, lactic acid (**A**), propanoic acid (**B**), erythritol (**C**), and 2,3-butanediol (**D**) were detected in all vinegar samples. The interquartile range (boxes), largest and smallest values (error bars), and medians (horizontal lines in the boxes) are indicated for each compound.

**Table 1 metabolites-11-00478-t001:** Vinegar samples used in this study.

Traditional Vinegars (*)	Geographical Origin in Korea	Raw Materials	Acetate (g/L)	Ethanol (g/L)	Glucose + Fructose (g/L)
TV1 (101)	Yeongdong, Chungbuk	Persimmon	37.6 ± 0.2	13.7 ± 0.2	0
TV2 (746)	Icheon, Gyeonggi	Persimmon	62.4 ± 1.6	1.7 ± 0.1	0.3 ± 0.03
TV3 (163)	Jinju, Gyeongnam	Persimmon	50.3 ± 0.1	2.5 ± 0.1	0
TV4 (553)	Wanju, Jeonbuk	Persimmon	35.2 ± 0.1	13.1 ± 0.3	0
TV5 (135)	Jeongeup, Jeonbuk	Persimmon	41.1 ± 1.7	12.7 ± 0.7	46.5 ± 2.2
TV6 (794)	Yeongcheon, Gyeongbuk	Brown rice	42.8 ± 1.5	0.4 ± 0.5	30.7 ± 1.1
TV7 (378)	Yecheon, Gyeongbuk	Multigrains	53.4 ± 0.7	4.6 ± 0.1	2.2 ± 0.3
**Commercial Vinegars**	**Manufacturer**	**Raw Materials**	**Acetate (g/L)**	**Ethanol (g/L)**	**Glucose + Fructose (g/L)**
CV1	Ottogi	Apple	64.5 ± 2.0	1.4 ± 0.03	11.3 ± 0.4
CV2	Ottogi	plum	63.5 ± 1.2	1.4 ± 0.1	4.1 ± 0.01
CV3	Daesang	Apple	51.6 ± 0.4	1.0 ± 0.04	8.8 ± 0.1
CV4	Ottogi	Brown rice	65.9 ± 1.1	1.5 ± 0.1	3.1 ± 0.1

* Certification numbers (in parenthesis) issued by the Ministry of Agriculture, Food and Rural Affairs, Korea.

**Table 2 metabolites-11-00478-t002:** Number of vinegar metabolites identified.

	Total Metabolites	Significantly Different Metabolites *	High in Traditional Vinegars ^a^	High in Commercial Vinegars ^b^
Volatile	102	5	4 (3)	1 (1)
Nonvolatile	78	15	14 (2)	1 (0)
Total	180	20	18 (5)	2 (1)

* VIP score > 1.0 and *p*-value < 0.05. ^a^ The numbers in parentheses refer to the number of TV-only compounds. ^b^ The numbers in parentheses refer to the number of CV-only compounds.

## Data Availability

The data presented in this study are available in supplementary tables.

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
