# Peer review of "Comparative Metabolite Profiling of Traditional and Commercial Vinegars in Korea"

_metabolites, 2021, doi:10.3390/metabo11080478_

Round 1

Reviewer 1 Report

This paper describes a comparative metabolite profiling of traditional and commercial vinegars in Korea. The article is quite complete and well thought out. The article is of interest to the scientific community and well discussed but I believe that it is necessary to improve several parts of the article.

In any case, I consider that the article is appropriate to be published in Metabolites journal once the authors apply the comments indicated below.

Include a table in the text with each of the 172 identified compounds, indicating whether they are common, specific or mostly specific for special-commercial vinegars.

Discuss whether there are volatile or non-volatile compounds, within these 172 compounds, specific to the origin of these vinegars (persimmon, rice, apple ...).

Material and methods: Include a reagent section

Minor comments

Lines 63, 69, 92, 115, 123,…: Put “p” and “t” in italics. Unify and apply to the entire document.

Section 3.2: Include the range of concentration studied for each compound.

Lines 184, 186, 192, 200,….: Include a separation between a number and “ºC”. Unify and apply to the entire document.

Lines 194, 202: Put “m/z” in italics. Unify and apply to the entire document.

References: Put the references in the correct format of the journal.

Author Response

Reviewer #1:

This paper describes a comparative metabolite profiling of traditional and commercial vinegars in Korea. The article is quite complete and well thought out. The article is of interest to the scientific community and well discussed but I believe that it is necessary to improve several parts of the article. In any case, I consider that the article is appropriate to be published in Metabolites journal once the authors apply the comments indicated below.

Response: We sincerely appreciate the reviewer for thoroughly evaluating the manuscript.

Include a table in the text with each of the 172 identified compounds, indicating whether they are common, specific or mostly specific for special-commercial vinegars.

Response: We thank the reviewer for the valuable suggestion. All identified compounds are listed in a new Table S1, and common and specific compounds were highlighted.

Discuss whether there are volatile or non-volatile compounds, within these 172 compounds, specific to the origin of these vinegars (persimmon, rice, apple ...).

Response: We thank the reviewer for the valuable suggestion. The following sentences highlight metabolites that are specific to raw materials:

Page 4, Lines 115-117: Meanwhile, there were 51 other TV-only compounds, 43 volatile and 8 nonvolatile compounds, with a low significance (p > 0.05) due to sample variations, which were in-dependent from raw materials (Table S12).

Page 4, Lines 130-134: There were 19 other CV-only volatile compounds with a low significance (p > 0.05) due to sample variations (Table S12). The unique compounds of each sample might be associated with the different ingredients, such as brown rice, apple, and pineapple, of the CVs. For example, methyl 5-hexenoate, a unique volatile compound of pineapple [28], was detected only in CV2, a pineapple-based commercial vinegar.

Material and methods: Include a reagent section

Response: We thank the reviewer for the suggestion. All reagents used in this study were listed in Sections 3.2 and 3.3, and their detailed information has been updated.

Minor comments

Lines 63, 69, 92, 115, 123,…: Put “p” and “t” in italics. Unify and apply to the entire document.

Response: According to the reviewer’s suggestion, “p” and “t” were changed to italic in the revised manuscript.

Section 3.2: Include the range of concentration studied for each compound.

Response: According to the reviewer’s suggestion, the following sentence was added to Section 3.2.

Page 6, Lines 181-183: The dynamic range of the standard curves was 0.1-5 g/L, and samples were appropriately diluted prior to analysis.

Lines 184, 186, 192, 200,….: Include a separation between a number and “ºC”. Unify and apply to the entire document.

Response: The entire manuscript has been revised according to the reviewer’s suggestion.

Lines 194, 202: Put “m/z” in italics. Unify and apply to the entire document.

Response: The entire manuscript has been revised according to the reviewer’s suggestion.

References: Put the references in the correct format of the journal.

Response: The references have been reformatted based on the journal guideline.

Reviewer 2 Report

The manuscript entitled “Comparative metabolite profiling of traditional and commercial vinegars in Korea” performed the metabolite profiling of traditional and commercial vinegar of Korea using gas chromatography – mass spectrometry. In my opinion, this article should be reconsidered for publication after major revisions, since there are less positive points. In addition, few samples were used in the present study, which allows obtaining an accurate conclusion.

Less positive points:

For example, in the abstract, a total of 102 nonvolatile and 70 volatile compounds were detected, but in the results and discussion section 102 volatile and 70 non volatile compounds were detected. Contradictory information!!!! The authors should present a table with volatile compounds identified, and another with non-volatile compounds. Moreover, the authors should provide information regarding how these compounds were identified.

Results and discussion: Table 1 is inserted in a section of metabolomic differences, when it only reports acetate, ethanol and glucose+fructose contents. I don’t understand the importance of this table in the current section.

Section Material and methods: Which conditions were used in the SPME procedure? Were optimized or based on a previous study?

Sometimes the authors used GC/MS others GC-MS.

Author Response

Reviewer #2:

The manuscript entitled “Comparative metabolite profiling of traditional and commercial vinegars in Korea” performed the metabolite profiling of traditional and commercial vinegar of Korea using gas chromatography – mass spectrometry. In my opinion, this article should be reconsidered for publication after major revisions, since there are less positive points. In addition, few samples were used in the present study, which allows obtaining an accurate conclusion.

Response: We sincerely appreciate the reviewer for thoroughly evaluating the manuscript.

Less positive points:

For example, in the abstract, a total of 102 nonvolatile and 70 volatile compounds were detected, but in the results and discussion section 102 volatile and 70 non volatile compounds were detected. Contradictory information!!!!

Response: We sincerely apologize for the confusion caused by the typos. The errors have been corrected in the revised manuscript.

The authors should present a table with volatile compounds identified, and another with non-volatile compounds.

Response: We thank the reviewer for the valuable suggestion. All identified compounds are listed in a new Table S1, and common and specific compounds were highlighted.

Moreover, the authors should provide information regarding how these compounds were identified.

Response: According to the reviewer’s suggestion, the following sentence was added to Section 3.4.

Page 7, Lines 207-208: For identification of the metabolites, Automated Mass spectral Deconvolution and Identification System (AMDIS) software, as described previously [36,37].

Results and discussion: Table 1 is inserted in a section of metabolomic differences, when it only reports acetate, ethanol and glucose+fructose contents. I don’t understand the importance of this table in the current section.

Response: According to the reviewer’s suggestion, Table 1 was relocated between Introduction and Results

Section Material and methods: Which conditions were used in the SPME procedure? Were optimized or based on a previous study?

Response: SPME procedure was based on a previous study.

Page 7, Lines 191-194: For volatile compounds, 5 mL of samples were extracted by SPME fiber coated with dinylbenzene/carboxen/polydimethylsiloxane (Bellefonte, PA, USA) at 50 °C for 60 min, as previously described [35]. The samples were injected by a MPS autosampler (Gerstel, Muelheim, Germany)…

Sometimes the authors used GC/MS others GC-MS.

Response: They were corrected to GC/MS for consistency.

Round 2

Reviewer 1 Report

The authors have made the indicated modifications and the article has improved substantially. For this reason, I consider that the article can be considered for publication in Metabolites journal in its current  form.

Author Response

We sincerely thank the reviewer for taking time and effort to review this manuscript.

Reviewer 2 Report

The manuscript entitled “Comparative Metabolite Profiling of Traditional and Commercial Vinegars in Korea” should be accepted since the authors included all reviewer’s suggestions. However, I continue to believe that this study includes few samples.

Correction:

“A total of 102 volatile and 78 nonvolatile compounds …” So, 180 metabolites, right?  Line 70: 172 metabolites??

Author Response

The manuscript entitled “Comparative Metabolite Profiling of Traditional and Commercial Vinegars in Korea” should be accepted since the authors included all reviewer’s suggestions. However, I continue to believe that this study includes few samples.

Correction:

“A total of 102 volatile and 78 nonvolatile compounds …” So, 180 metabolites, right?  Line 70: 172 metabolites??

Response: We sincerely thank the reviewer for taking time and efforts to review our manuscript, and finding a critical typo after the first revision. The re-revised manuscript has been fixed as follows:

Lines 70, 285: 172 > 180

This manuscript is a resubmission of an earlier submission. The following is a list of the peer review reports and author responses from that submission.